# A Formalized Method to Acclimate Dogs to Voluntary Treadmill Locomotion at Various Speeds and Inclines

**DOI:** 10.3390/ani12050567

**Published:** 2022-02-24

**Authors:** Alexander R. Stigall, Brian D. Farr, Meghan T. Ramos, Cynthia M. Otto

**Affiliations:** 1Penn Vet Working Dog Center, Department of Clinical Sciences and Advanced Medicine, School of Veterinary Medicine, University of Pennsylvania, Philadelphia, PA 19146, USA; megramos@vet.upenn.edu (M.T.R.); cmotto@vet.upenn.edu (C.M.O.); 2Army Medical Department Student Detachment, 187th Medical Battalion, 32nd Medical Brigade, Joint Base San Antonio—Fort Sam Houston, San Antonio, TX 78234, USA

**Keywords:** dog, exercise, protocol, locomotion, behavior, low stress, soreness

## Abstract

**Simple Summary:**

The land treadmill provides a range of behavioral and physical training benefits for dogs. Walking and trotting on the treadmill, however, is unfamiliar to many dogs and requires acclimation. This study developed and conducted a voluntary treadmill acclimation protocol on eight working dogs in training or working dogs performing detection research. The acclimation protocol was successfully completed for seven out of eight dogs. An acclimation assessment protocol was developed to measure a previously exposed dog’s degree of acclimation. This protocol was successfully used in two previously exposed dogs. A muscle soreness protocol was created to evaluate the soreness developed during the acclimation protocol. These protocols offer an option to acclimate dogs to the treadmill and determine the degree of acclimation for previously exposed dogs for research and training purposes.

**Abstract:**

The land treadmill is a multipurpose tool with a unique set of behavioral and physical benefits for training and assessing active dogs. Habituation to voluntary treadmill locomotion is crucial for training a dog or accurately assessing a dog’s fitness on a treadmill. Therefore, a treadmill acclimation program was developed and evaluated with working dogs in training or working dogs performing detection research. Seven of eight naive dogs became acclimated to the treadmill using the protocol developed. Two previously experienced dogs successfully conducted an acclimation assessment to test for habituation to the treadmill. A muscle soreness protocol was created to evaluate the soreness developed during the acclimation program. This detailed protocol was successful in acclimating dogs to the treadmill at various safe speeds and inclines.

## 1. Introduction

Locomotion, specifically aerobic exercise, provides benefits for the health of dogs because it develops cardiopulmonary and muscular endurance, is behaviorally stimulating, and can lead to weight loss [1]. The common methods to provide aerobic exercise for dogs include walking, hiking, or running on-leash, retrieving objects, swimming, and both underwater and land treadmills. Although the land treadmill does not replicate the exact locomotion pattern of running [2,3,4], the land treadmill provides unique benefits compared to other aerobic exercise methods. The dog can be exercised in a structured and controlled manner that is convenient for the handler/owner with limited influences by environmental or climate effects [5,6]. The treadmill also can be a tool for fitness research [6,7,8,9,10,11,12,13,14,15,16], diagnostic gait analysis, and rehabilitation treatment.

The land treadmill is typically unfamiliar to most dogs and therefore requires behavioral acclimation before it can be effectively used. There are many challenges associated with acclimating a dog to the treadmill. A dog must become comfortable with the treadmill belt moving below their feet, the speed and incline changing, maintaining their position on the running surface, and remaining on the treadmill for extended periods of time. Acclimation makes a dog more comfortable and motivated to run on the treadmill [14,16]. Failure to acclimate appropriately can result in anxiety, aversion, injury, or limited performance [14,17].

Although the land treadmill has been utilized in many scientific studies (see references) and is commonly recommended in fitness programs [6,12], few studies have reported detailed protocols describing how dogs were behaviorally acclimated to the treadmill [8]. Many treadmill studies use forced exercise protocols [7,12] where the dog is typically leashed to the treadmill and is prevented from stopping by the researchers. In order to safely and effectively introduce dogs to the treadmill while being mindful of their welfare, a formalized method to acclimate dogs to voluntary treadmill locomotion is needed. A method to determine the degree of acclimation of previously exposed dogs is also needed. In this study, we developed a formalized protocol to acclimate dogs to voluntary treadmill locomotion at various speeds and inclines and an acclimation protocol for previously exposed dogs.

## 2. Materials and Methods

### 2.1. Experimental Setup

#### 2.1.1. Equipment

A motorized treadmill (Large DogTread treadmill, Petzen, Ogden, UT, USA) was used for all acclimation sessions. Acclimation sessions were conducted in a room that was climate controlled (21.7–24.4 °C (71–76 °F), 56–61% humidity), free of distractions, and familiar to all dogs. The treadmill was positioned with ample space behind it to allow dogs to walk on and off the treadmill. The treadmill inclines (2%, 10%, and 20%) corresponded with a subsequent treadmill assessment study. The 2% incline was obtained by placing the front of the treadmill on the ground without using the stand at the end (Figure 1A). The 10% incline was obtained by placing the front of the treadmill 10 cm off the ground (on two rows of bricks each measuring 5 cm high) (Figure 1B). The 20% incline was obtained by placing the front of the treadmill 25 cm off the ground (on a row of concrete blocks (measuring 20 cm high) and a row of bricks (measuring 5 cm high)) (Figure 1C). The object height to obtain each incline was determined using a smartphone application (‘Measure’ by Apple Inc., Cupertino, CA, USA). A towel was placed between the bricks or blocks and the treadmill to prevent movement and decrease vibration (Figure 1D). A 10 cm by 10 cm wood beam was placed across the belt at the front of the treadmill to prevent dogs from accidentally stepping on the nonmoving part of the treadmill (Figure 1D).

#### 2.1.2. Reward Types

The reward options for dogs were food and toys. Food rewards were initially used for all dogs, as they allowed intermittent rewards during the acclimation session. A licking-type food reward (frozen peanut butter and water) was used to ease swallowing while moving on the treadmill. If any dog was not motivated for food, a toy reward was used and was given to the dog after increasing intervals of locomotion on the treadmill.

### 2.2. Experiment 1—Treadmill Acclimation Protocol

#### 2.2.1. Participants

All participants were either working dogs in training or working dogs performing detection research at the Penn Vet Working Dog Center (PVWDC, Philadelphia, PA, USA). Dogs conducted their normal training in addition to the study. All dogs received a functional musculoskeletal examination from one of the study veterinarians. The trainers of all dogs provided information about each dog’s amount, quality, and currency of prior treadmill experience.

To be included in the study, a dog must have been eight months of age or older, free from performance-limiting disease, present for the entire study, and had not been previously acclimated to trotting on the treadmill.

Dogs were excluded from the study if they were under eight months old, had a performance-limiting issue, were not present for the entire study, and/or were previously acclimated to being behaviorally comfortable trotting on the treadmill.

#### 2.2.2. Procedure

##### Protocol

Dogs conducted their normal training during the study, consisting of approximately two hours of activity per day. Acclimation sessions were scheduled either before or after this training with the goal being to acclimate each dog during a consistent time of day. The acclimation protocol (Appendix A) consisted of a series of levels and criteria for progressing. The levels incrementally increased the treadmill speed from 0.5 mph (0.8 kph) to 5.0 mph (8.0 kph) at 2%, 10%, and 20% inclines. The maximum speed and incline were chosen to acclimate dogs for a subsequent treadmill assessment study. A dog was considered fully behaviorally acclimated when they were either able to complete all levels in the protocol (e.g., 5.0 mph (8.0 kph) at 20% incline for 30 s) or were physically unable to complete a level despite being behaviorally comfortable. The protocol has since been revised to a maximum speed of 7.0 mph (11.3 kph) with the same inclines.

##### Acclimation Session Procedure

Each treadmill acclimation session followed a standard procedure (Appendix A) and was conducted by a single dog handler (A.R.S). The dog was assessed for muscle soreness and performed the PVWDC Fit to Work (FTW) Warm-up [18]. If the dog was comfortable walking on the treadmill, they would perform an on-treadmill warm-up at a walking pace. The dog would begin their acclimation progression two levels below what they reached in the previous session. If the dog was comfortable at this level, the handler would advance the dog to a new level. If the dog was comfortable with the novel level, the handler would increase the speed and/or incline to let the dog move to the next level. If a dog was observed to be behaviorally stressed (e.g., looking to the side, panting unrelated to physiological stress, disengagement from the reward unrelated to physiological stress, or caudal ear position), the handler would decrease the speed and/or incline to where the dog was comfortable before attempting to make the progression again. The session ended if the dog was either consistently uncomfortable due to physical or behavioral stress, or because they had completed a predetermined duration (10 min) of treadmill exposure. Sessions were limited to 10 min to increase the likelihood of ending with a positive experience, rather than extending the time of the session and potentially creating a negative experience for the dog. If the dog was comfortable walking on the treadmill, they would perform an on-treadmill cool-down at a walking pace. The dog would then get off the treadmill and perform the PVWDC FTW Cool-down [18]. The goal was for each dog to complete three acclimation sessions per week.

##### Muscle Soreness Assessment

A standardized method was developed to assess muscle soreness and reluctance to perform functional movements. The method was developed by the veterinarians involved in the study (B.D.F., M.T.R. and C.M.O.) and taught to the handler (A.R.S.). The muscle soreness assessment was performed by the handler (A.R.S.) on each dog before every acclimation session. The muscle soreness assessment was not utilized prior to Sessions 1–3 in all dogs but was introduced after post-training soreness was noticed in some dogs (Sessions 4–7).

The muscle soreness assessment consisted of three components: scanning muscle groups for heat, palpating for muscular tension while observing for behavioral response, and observing functional performance during the PVWDC FTW Warm-up. The muscle groups assessed were the epaxials, gluteals, cranial thighs, and caudal thighs, as well as the individual iliopsoas muscle. All muscle groups were assessed bilaterally. Any heat in a particular muscle group was noted and recorded as present or absent. Muscular tension was graded as no tension (0), mild tension (1), or moderate or greater tension (2). The muscular tension scores for each muscle group were added to obtain a total muscular tension score. Any repeatable reluctance to perform the PVWDC FTW Warm-up movements was noted and recorded as present or absent. The results of the muscle soreness assessment were recorded on a standardized form (Appendix A).

If a dog had palpable heat in any muscle group, a muscular tension score of 2 in any muscle group, a total muscular tension score of 4 or greater (e.g., a score of 1 in 4 different groups), or reluctance to perform any of the PVWDC FTW Warm-up movements, they did not participate in the planned acclimation session. The handler notified a veterinarian involved in the study, completed the PVWDC FTW Warm-up, performed a 5 min walk (if acclimated to at least level 5) or trot (if acclimated to at least level 17) on the treadmill with no additional incline, and completed the PVWDC FTW Cool-down. A veterinarian involved in the study then evaluated the dog, prescribed the appropriate treatment, and reevaluated the dog as appropriate. The muscle soreness was categorized into three groups: acute muscle soreness (<24 h post-acclimation session), delayed onset muscle soreness (24 h–5 days post-acclimation session), and prolonged muscle soreness (>5 days post-acclimation session) [19]. The dog was assessed for residual muscle soreness at the next treadmill acclimation session.

### 2.3. Experiment 2—Treadmill Acclimation Assessment

#### 2.3.1. Participants

Participants had the same inclusion criteria and exclusion criteria as Experiment 1 participants, except they were required to be behaviorally comfortable trotting on the treadmill.

#### 2.3.2. Acclimation Assessment

These dogs performed an assessment of their familiarity with various speeds and inclines of the treadmill acclimation protocol. The acclimation assessment followed a standard procedure (Appendix A) and was conducted by a single handler (A.R.S). The dog first performed the PVWDC FTW Warm-up [18] and then walked on the treadmill at 2.0 mph (3.2 kph) for 2 min on the 2% incline (flat) treadmill. The speed of the treadmill increased by 1.0 mph (1.6 kph) every 30 s to a maximum speed of 7.0 mph (11.3 kph) or until the dog demonstrated they were physically unable to move at that pace. The treadmill was stopped, and the dog rested for 1 min. The dog repeated the process on the 10% and 20% incline treadmill. The assessment ended if a dog was consistently uncomfortable due to physical or behavioral stress. The dog performed a cool-down on the treadmill by walking at 1.5 mph (2.4 kph) for 2 min. The dog then got off the treadmill and performed the PVWDC FTW Cool-down [18]. The dog was considered fully acclimated if either they completed the acclimation assessment or were physically unable to complete an earlier progression despite being behaviorally comfortable at that level.

### 2.4. Statistics

The data were evaluated for normality using the Shapiro–Wilk test [20]. If the data were normally distributed, the mean and standard deviation (SD) were reported. If the data were not normally distributed, the median and range of nonparametric data were reported.

## 3. Results

### 3.1. Participants

Eight dogs (one search and rescue, three law enforcement, two detection research, and two with undetermined careers) participated in Experiment 1 (Table 1). At the start of the treadmill acclimation process, they had a median age of 1.44 years (range of 0.71–7.22 years) and a mean body condition score of 4.0/9.0 (+/−0.8). Two dogs (both detection research careers) participated in Experiment 2 (Table 1). They were both adults and had a body condition score between 4.5 and 5.0/9.0 when they conducted the treadmill acclimation assessment.

### 3.2. Experimental Setup

#### Reward Types

A licking food reward was used to reward all but one dog. All dogs that received a licking food reward used a frozen peanut butter and water mix. One dog (Gunner) was not motivated to walk or trot on the treadmill for food and was therefore rewarded with a toy. Gunner would walk or trot on the treadmill for increasing durations with the goal of rewarding at the end of each level.

### 3.3. Protocol

#### 3.3.1. Scheduling of Participants

Dogs were scheduled around their normal career training. Four dogs (Bobbie, DJ, Osa, and Ross) consistently performed acclimation sessions in the morning before normal training, and four dogs (Dozer, Fury, Gunner, and Sheridan) performed sessions in the afternoon after normal training. Nearly all (64/70, 91%) sessions were performed at a consistent time of day for each dog. The median number of acclimation sessions per dog per week was 2 (range of 0–4 sessions per dog per week). The median number of days between sessions was 2 (range of 1–20 days). Some dogs had extended periods without any acclimation sessions due to absence from the training facility or treatment for medical issues (see ‘Physiological Effects’).

#### 3.3.2. Protocol Progression

Seven of the eight dogs (88%) were fully behaviorally acclimated to the treadmill to the limit of their physical ability. The acclimation protocol was performed in preparation for a subsequent treadmill assessment study, which required the dogs to become habituated to a maximum speed of 5.0 mph (8.0 kph) and 20% incline (level 23). Four dogs (Dozer, Fury, Gunner, and Ross) completed the protocol at level 23 in preparation for this study, two dogs (DJ and Sheridan) were physically limited at level 16, and 1 dog (Bobbie) was physically limited at level 12 (Figure 2).

The mean number of sessions required to acclimate a dog, excluding the one dog who was not fully acclimated, was 9.4 (+/−1.9). The median number of levels completed per session was 2.0 (range of −7.0–11). The majority (57/70, 81%) of sessions resulted in an increased progression. 9/70 sessions (13%) resulted in no change in progression, and 4/70 sessions (6%) resulted in a decreased progression. Of the 13 sessions that did not end in an increased progression, 10 sessions were due to behavioral stress or unwillingness to perform, 1 session was due to physical limitation, and 2 sessions were unclear as to whether behavioral or physical stress prevented progression.

All fully acclimated dogs completed the protocol within a 5-week period (Figure 2). The mean number of days required to reach full acclimation (including nontreadmill days) was 29 days (+/−7.0), excluding Osa, who did not become fully acclimated. Bobbie and Ross had a two-week and one-week absence from the training facility, respectively. Both DJ and Dozer had one-week breaks from the treadmill as a result of long bone pain (see ‘Physiological Effects’).

One dog (Osa) was unable to acclimate to the treadmill (Figure 2). Despite multiple attempts to positively reinforce treadmill acclimation, Osa showed consistent signs of behavioral stress, which may have been confounded by physical soreness (see ‘Physiological Effects’), and the protocol was ended.

#### 3.3.3. Treadmill Acclimation Assessment

Both Experiment 2 dogs (Ivey and Toby) completed the treadmill acclimation assessment at speeds up to 5.0 mph (8.0 kph) on the 2%, 10%, and 20% incline. These dogs conducted the acclimation assessment in preparation for a subsequent treadmill assessment study, which required them to be acclimated to a maximum speed of 5.0 mph (8.0 kph).

#### 3.3.4. Physiologic Effects

Six dogs (DJ, Fury, Gunner, Osa, Ross, and Sheridan) developed 11 instances of muscle soreness during the protocol (Figure 2). Each of these dogs had at least one episode of muscle soreness. DJ, Ross, and Sheridan each had two episodes of muscle soreness. Osa had three episodes of muscle soreness. The mean soreness score of instances of muscle soreness was 2 (range of 2–7). Two of the Experiment 1 dogs (Dozer and Bobbie) and both Experiment 2 dogs did not have any episodes of muscle soreness.

If a dog’s muscle soreness score was between 1 and 4, the dog’s trainer was asked to report any changes in the dog’s performance in their normal training. If a dog’s muscle soreness score was above 4, they were treated for their muscle soreness. The muscle soreness treatment protocol consisted of nonsteroidal anti-inflammatory drugs (NSAIDs), therapeutic massage, and photobiomodulation therapy (Luminex Vet Class IIIb laser, Respond Systems Inc., Branford, CT, USA). Photobiomodulation therapy was only included in the treatment protocol if the dog was greater than 12 months of age. Additional treatment or activity restrictions were implemented by the veterinarians for specific episodes of soreness and are described below.

Acute muscle soreness was identified in four dogs (Gunner, Osa, Ross, and Sheridan). Gunner developed bilateral iliopsoas soreness within 30 min after his last treadmill session. He had a muscle soreness score of 2, as identified by fasciculations. The veterinarians did not provide any additional treatment. No soreness was noticed in Gunner in the days following this treadmill session and for the remainder of the study. Osa developed bilateral iliopsoas soreness within 30 min after her last treadmill session. When this soreness was identified, her trainer reported she went swimming for multiple hours 2–3 days prior. It is possible the swimming caused or predisposed her to the soreness. She had a muscle soreness score of 4 and was reassessed seven days later. Her soreness did not resolve by the next soreness assessment. Therefore, her soreness was categorized as prolonged muscle soreness and specified further below. Ross developed bilateral iliopsoas within 30 min after his last treadmill session. He had a muscle soreness score of 2 as identified by unwillingness to perform an iliopsoas stretch during the cool-down. The veterinarians did not provide any additional treatment. His soreness did not resolve by his next soreness assessment 24 h later. Therefore, his soreness was categorized as delayed onset muscle soreness and specified in the paragraph below. Sheridan developed bilateral iliopsoas soreness within two hours after his last treadmill session and had a muscle soreness score of 2. The veterinarians put him on an activity restriction for the weekend (no activity besides light walks), and he was reassessed three days later. His soreness did not resolve by the next soreness assessment. Therefore, his soreness was categorized as delayed onset muscle soreness and specified in the paragraph below.

Delayed onset muscle soreness was identified in four dogs (DJ, Fury, Ross, and Sheridan). DJ developed bilateral epaxial soreness four days after his last treadmill session. He had a muscle soreness score of 2, as identified by muscle tightness. DJ did not receive any additional treatment for this soreness and completed a treadmill session the same day. The next day, DJ developed cranial thigh soreness. He had a muscle soreness score of 2, as identified by muscle warmth. DJ did not receive any additional treatment for this soreness. No muscle soreness was noticed in DJ in the days following this treadmill session and for the remainder of the study. Fury developed bilateral gluteal soreness four days after his last treadmill session. He had a muscle soreness score of 2, as identified by muscle warmth. Fury did not receive additional treatment. No soreness was noticed in Fury in the days following this treadmill session and for the remainder of the study. Ross’s soreness was identified within 30 min after his last treadmill session but persisted for the following day. Bilateral iliopsoas soreness was noticed as a result of unwillingness to participate in an iliopsoas stretch and he received a muscle soreness score of 2. No muscle soreness was noticed in Ross for the remainder of the study. Sheridan’s soreness was identified within two hours after his last treadmill session but worsened three days later (muscle soreness score of 7). A PVWDC trainer noticed poor performance from Sheridan in his normal training, and bilateral iliopsoas, right cranial thigh, and bilateral caudal thigh soreness was noticed from a physical examination from a veterinarian. As a result of this muscle soreness score, the treatment protocol was provided. Sheridan was given NSAIDs for five days and an effleurage massage and photobiomodulation therapy immediately after the physical examination. No muscle soreness was noticed in Sheridan for the remainder of the study.

Prolonged muscle soreness was identified in one dog (Osa). Osa’s bilateral iliopsoas soreness was reassessed six days after her last treadmill session when her soreness began. Osa’s bilateral iliopsoas soreness persisted, and she was unwilling to perform the warm-up of the treadmill session. She had a muscle soreness score of 5. As a result of this muscle soreness score, the treatment protocol was provided. Osa was given NSAIDs for seven days and an effleurage massage and photobiomodulation therapy immediately after the muscle soreness assessment. Osa was reassessed for soreness two days later. Her bilateral iliopsoas soreness persisted, and she received a muscle soreness score of 4. She received an effleurage massage and photobiomodulation therapy immediately after the muscle soreness assessment. Osa’s acclimation protocol was ended that day in the interest of her health and well-being.

In addition to the muscle soreness, long bone pain was identified on palpation in two young dogs after lameness was noticed (DJ and Dozer). DJ had two incidents of long bone pain on the same bone six to eight weeks prior to the acclimation protocol. Dozer had no prior incidents of long bone pain. DJ and Dozer were treated with NSAIDs for 7 days and placed in an activity restriction. DJ and Dozer were restricted from incline treadmill work, jumping, ball retrieval, and social play with other dogs for the 7-day period. They were reassessed by a veterinarian following the initial 7 days and had an additional 7 days of NSAIDs and activity restriction after long bone pain was identified in both dogs.

DJ’s long bone pain was identified the same day as his last treadmill session. He was provided the treatment protocol until his long bone pain was resolved. His NSAID therapy and activity restriction lasted 14 days. Dozer’s long bone pain was identified six days after his last treadmill session. He was provided the treatment protocol until his long bone pain was resolved. His NSAID therapy and activity restriction lasted 14 days. Dozer also performed one treadmill walk session for active recovery (this data point is not shown in Figure 2 because Dozer was already acclimated to walking on the treadmill). There have been no recurrences of long bone pain in 11 weeks since the end of the treadmill acclimation protocol.

## 4. Discussion

The purpose of this study was to develop a voluntary treadmill exercise acclimation protocol for active dogs. Seven of the eight naive dogs became fully behaviorally acclimated to the treadmill to their physical ability using this protocol. Both previously exposed dogs successfully completed the treadmill acclimation assessment as a method to evaluate their ability to comfortably perform treadmill exercise of increasing difficulty. A formalized treadmill acclimation protocol, session procedure, simplified muscle soreness assessment, and treadmill assessment were developed in this study.

The study utilized a diverse group of working dogs. This diversity, based on current or future careers, age, and breed, provides support for the treadmill acclimation protocol’s effectiveness in a range of working dogs. All dogs that completed the protocol were fully acclimated in 5 weeks, with a median of two acclimation sessions per dog per week. Four dogs performed acclimation sessions in the morning prior to normal training, and four dogs performed acclimation sessions in the afternoon after normal training.

The treadmill has been used in research studies across many species, including dogs, horses, rodents, and humans. Many studies have conducted treadmill exercise tests in dogs [6,7,8,9,10,11,12,13,14,15,16]. Some studies used forced exercise (i.e., no acclimation conducted) protocols [7,12], while others have used acclimation protocols [6,8,9,10,11,13,14,15,16]. These studies typically report dogs were acclimated to the treadmill but do not provide specific details on acclimation protocols. There is little to no mention of the timeline, previous familiarization to the treadmill, number of participants removed due to inability to acclimate, development of muscle soreness, and session-by-session procedures for acclimating dogs [6,8,9,10,11,13,14,16,17]. Some studies also report short acclimation protocols (e.g., two acclimation sessions), which brings to question if the subjects were fully behaviorally acclimated to, and comfortable on, the treadmill [7,16]. Some studies have researched the duration (e.g., for 10–20 min) of treadmill locomotion required to physiologically habituate dogs to the treadmill directly before conducting gait analysis [4,5,21]. These studies also did not require a dog to trot at more intense inclines and speeds [4,5,21]. To use the treadmill as a long-term training or research tool, a dog must be behaviorally acclimated and motivated to walk or trot on the treadmill. To the authors’ knowledge, there are no prior studies conducting and evaluating a voluntary treadmill acclimation protocol.

While treadmills are used to assess the performance of horses, little research on behavioral acclimation of horses to the treadmill has been reported [22]. Masko et al. created an ethogram to evaluate the behavioral acclimation of horses to treadmill locomotion [22]. Horse treadmill exercise studies with acclimation protocols do not provide the details of the protocol [22,23,24,25]. Similar to dog studies, the acclimation required to physiologically habituate horses to treadmill locomotion for gait analysis and accurate testing measurements has been investigated [24]. Many treadmill protocol studies in rodents [26,27,28,29,30,31] use forced exercise protocols rather than voluntary protocols requiring acclimation [26,27,28,29,30]. In humans, acclimation protocols are typically not required, as it is expected in most studies humans have used or can quickly learn the skill of running on the treadmill. One study conducted a treadmill walking program for adolescents with autism spectrum disorder but did not include any details on acclimating subjects to treadmill walking [32].

One dog (Bobbie) with previous negative experience on the treadmill was fully acclimated to the treadmill using this protocol. Bobbie’s acclimation progression followed a similar curve to the other fully acclimated dogs (Figure 2). The median number of levels completed per session was 2.0 (range of −7.0–11), which supports this approach was effective in continually progressing dogs throughout the protocol. The protocol was efficient with a mean of 9.4 sessions (+/−1.9) required to fully acclimate a dog (excluding Osa, who was not fully acclimated). All fully acclimated dogs had similar progressions: early in the training, moving through levels 1–7 in the first 3–4 sessions. The timeline to full acclimation was similar across all fully acclimated dogs. Thirteen of the 70 total acclimation sessions across all dogs ended without completing a new level. Four of these thirteen sessions ended with a retrogression, and the remaining nine sessions ended with neither a progression nor retrogression (i.e., the dog completed the same level as their previous session). Most of these sessions were small decreases or no change due to behavioral or physical limitations. Dozer and Gunner both had one substantially poor (level 1) acclimation session on the same day (Figure 2). Both dogs were unwilling and unmotivated to walk on the treadmill on this particular day. Both dogs conduct normal training with the same trainer and were reported to have completed a relatively intensive day of training prior to this acclimation session. These dogs may have been too tired to effectively perform an acclimation session on this day. This observation supports scheduling acclimation sessions in the morning prior to normal training whenever possible.

In this protocol, both incline and speed were changed. The PVWDC trainers helped develop the progression protocol based on their previous experience acclimating dogs to the treadmill. The trainers predicted dogs would have a more difficult time adjusting to increasing speeds compared to increasing inclines during treadmill acclimation. Therefore, the progression protocol interspersed levels that increased speed with those that increased incline. This protocol attempted to balance the behavioral and physical challenges of increasing difficulty by increasing the incline at a set speed to acclimate a dog to a more intense level on the treadmill without becoming behaviorally challenging. The speed was increased on a lower incline so it would not be as physically demanding and would therefore not be as behaviorally challenging for the dog. The progression protocol was successful in continuously progressing dogs. Fifty-seven out of 70 acclimation sessions led to progression. This suggests the increments were small enough to allow dogs to progress in the acclimation protocol.

The maximum speed of the treadmill acclimation protocol was chosen for three reasons. First, a major priority of the study was the safety and welfare of the dogs. Higher speeds on the treadmill cause a dog to canter, increasing the risk of stumbling or falling off the treadmill. Second, another priority of the study was to create an acclimation protocol applicable to the average handler or owner. The maximum speed of many commercially available treadmills is 7.5 mph (12.1 kph), and this study sought to stay within that limitation. Finally, the protocol was originally limited to 5 mph (8.0 kph) at a 20% incline as the observable effort required by the dogs in the study to complete this level was sufficient for the associated treadmill assessment study. We have since revised the protocol to a maximum level of 7 mph (11.3 kph) at a 20% incline to accommodate the application of more fit dogs.

Seven of the eight dogs were fully acclimated to the treadmill (Figure 2). Four dogs completed the maximum stage (5.0 mph on 20% incline) and three dogs were physically limited before reaching the maximum stage (Figure 2). The three dogs that were physically limited (Bobbie, DJ, and Sheridan) have been assessed by PVWDC veterinarians (B.D.F., M.T.R., and C.M.O.) to lack physical fitness. Bobbie and Sheridan both had high body condition scores (both 5/9) and were relatively heavy (33.8 and 40.9 kg, respectively) compared to the other dogs in the study. The work required to travel is greater with a heavier load (i.e., heavier individuals require more work for locomotion on the treadmill) [33]. Longer limb length decreases the cost of energy required to travel [33]. It is possible the limited hip extension of German Shepherds contributed to the physical limit of Bobbie and Sheridan but cannot be the only reason for their limited performance, as the other German Shepherd, Dozer, was able to complete the maximum stage.

One dog (Osa) was unable to complete the acclimation protocol. Osa had previous negative experiences on the treadmill according to her trainer (Table 2). Osa did not show any major signs of behavioral or physical discomfort in her first three acclimation sessions. After her third session, bilateral iliopsoas soreness was identified. In her next session, Osa showed clear physical and behavioral signs of stress such as lameness, lethargy, displacement activity, and panting. Two months prior to the start of the acclimation protocol, Osa had developed an iliopsoas injury. Although she was medically cleared as healthy by the start of the acclimation protocol, the treadmill may have caused a re-injury or exacerbation of a subclinical condition. It is unclear if her behavioral stress was due to her iliopsoas discomfort, other behavioral stress, or a combination of both.

A session-by-session protocol was created to outline the details of an acclimation session while utilizing low-stress handling techniques to effectively familiarize a dog to the treadmill. For long-term treadmill use, acclimation may limit or prevent aversion. There is little detailed information on acclimation protocols using low-stress handling techniques for dogs. For research studies, it may also benefit using low-stress handling methods to prevent behavioral stress which affects physiological results. Some studies have published methods to acclimate rodents to the treadmill environment prior to forced exercise [26,27,28,30,31]. These studies measured various biochemical markers, such as plasma lactate and heart rate, which both can be affected by stress. A session-by-session protocol is also beneficial for working and sporting dogs because a detailed protocol can outline how to schedule acclimation sessions within their normal training.

The room used for all acclimation sessions was arranged to reduce distractions for the dogs. Nine of the 10 dogs used a food reward (frozen peanut butter and water mix). Food was rewarded intermittently and approximately every 15 s. One dog (Gunner) was unwilling to walk or trot on the treadmill for food. A toy reward was used for Gunner because it was more motivating for him. Gunner would complete increasing durations on the treadmill and was only rewarded after successfully completing the level. The treadmill was stopped before he was rewarded to prevent him from jumping for a ball on the moving treadmill. A food reward is favorable because it allows a high rate of reinforcement and allows the handler to reward intermittently to maintain the dog’s motivation to walk or trot on the treadmill. A toy reward requires more behavioral impulse control training.

Multiple canine pain assessments have been created [34,35,36,37]. Common pain assessments used are the Canine Brief Pain Inventory [34], Helsinki Chronic Pain Index [37], and Liverpool Osteoarthritis in Dogs questionnaire [35]. These pain assessments, however, do not specifically assess exercise-induced muscle soreness. Riley et al. used a myofascial and musculoskeletal pain scale to evaluate the efficacy of massage therapy [38]. This scale is similar to the muscle soreness assessment developed in this study. The perceived benefits of our soreness assessment include time efficiency, simplicity to learn and conduct, and low stress for the dog. The muscle soreness assessment was developed following the observation of soreness in multiple dogs in the early part of the study. Acute or delayed muscle soreness is a common adverse event following intensive activity. The muscle soreness protocol demonstrated that the most common adverse events were acute (<2 h) or delayed (2 h–5 days) muscle soreness. The muscle soreness was evaluated and characterized as palpable muscle fasciculations, pain, inhibited range of motion at the joint with change on end feel or restriction or myofascial trigger points. Future research in validating this soreness assessment with previously validated pain scales [34,35,36,37] would provide handlers and researchers with a simple and effective muscle soreness evaluation tool.

Long bone pain was identified in DJ and Dozer by palpation. DJ and Dozer also performed normal training during the acclimation protocol. It is inconclusive if treadmill acclimation sessions exacerbated or caused long bone pain, or if the larger volume of training, regardless of the activity exacerbated or caused long bone pain.

An acclimation assessment was created to assess the familiarity of the treadmill to previously exposed dogs. Ivey and Toby performed the acclimation assessment. Both dogs successfully completed the acclimation assessment to a maximum stage of 5.0 mph on the 20% incline. The muscle soreness assessment was not created at the time these dogs completed the acclimation assessment, but no subjective muscle soreness was reported in either dog immediately after or the days following their acclimation assessments.

One limitation of the study was the small sample size. Conducting the acclimation protocol on a larger and more diverse group of dogs would provide more information on the behavioral challenges associated with treadmill acclimation. It is possible a correlation may be found between acclimation plateaus and a specific level in the protocol. Another limitation of the study was no physiologic measures of stress were obtained, and no ethogram was created for a standardized evaluation of the behavioral stress associated with acclimation. Replicating the study and measuring stress biomarkers such as salivary cortisol or heart rate would quantify the stress induced by the acclimation program. Creating an ethogram to assess the behavioral stress signs during acclimation sessions may also show correlations between behaviors and specific levels in the protocol and be able to substantiate that this was a low-stress protocol. The muscle soreness score was not instituted from the beginning and was not a validated score. Comparing the muscle soreness assessment to previously validated canine pain assessments, and refining the methods to identify, track, and mitigate exercise-induced muscle soreness is a future direction.

## 5. Conclusions

A detailed treadmill acclimation protocol, session-by-session voluntary protocol, muscle soreness assessment, and acclimation assessment were developed. A group of active working dogs and working dogs in training were successfully acclimated to voluntary treadmill locomotion.

## Figures and Tables

**Figure 1 animals-12-00567-f001:**
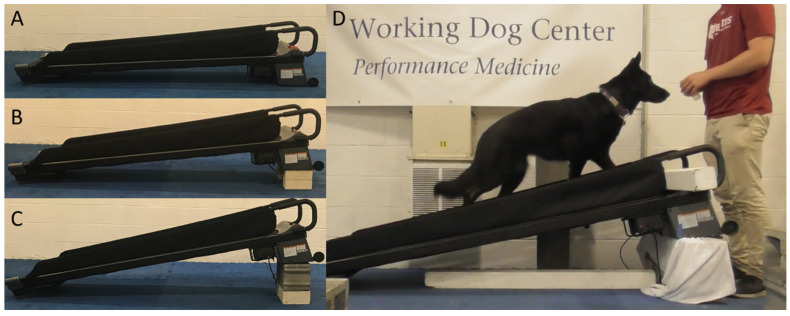
(**A**) 2% incline treadmill without elevation. (**B**) 10% incline treadmill elevated to 10 cm. (**C**) 20% incline treadmill elevated to 25 cm. (**D**) Dog performing an acclimation session on the full 20% incline treadmill setup.

**Figure 2 animals-12-00567-f002:**
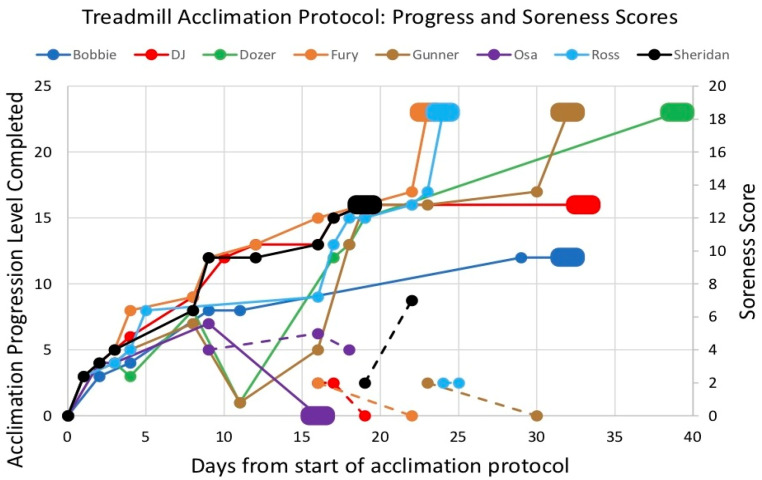
Treadmill acclimation levels and muscle soreness scores. The solid lines indicate the treadmill acclimation levels, and the dashed lines (same color) indicate the soreness scores for each dog. The only zero soreness score values plotted are ones directly following nonzero soreness score values to identify when soreness was identified as resolved.

**Table 1 animals-12-00567-t001:** Demographics of participants in Experiments 1 and 2.

Experiment	Name	Sex	Age (Years)	Breed	Body Condition Score (1.0–9.0)	Weight (kg)
	Bobbie	Spayed Female	5.40	German Shepherd	5.0	33.8
	DJ	Male	0.71	Dutch Shepherd	3.0	29.1
	Dozer	Male	0.73	German Shepherd	4.5	36.7
1	Fury	Male	1.81	Belgian Malinois	3.0	29.6
	Gunner	Male	1.34	Dutch Shepherd	3.5	29.6
	Osa	Spayed Female	7.22	German Shepherd	4.0	28.6
	Ross	Male	0.79	Labrador Retriever	4.0	29.2
	Sheridan	Male	1.53	German Shepherd	5.0	40.9
2	Ivey	Spayed Female	4.50	German Shepherd	4.5	29.3
	Toby	Neutered Male	2.99	Small Münsterländer	5.0	20.3

**Table 2 animals-12-00567-t002:** Physical and behavioral treadmill familiarity characteristics of participants in Experiments 1 and 2 based on the knowledge of the dogs’ trainers.

Experiment	Name	Prior Treadmill Experience	Quality of Prior Experience	Amount of Prior Experience	Currency of Prior Experience
	Bobbie	Yes	Negative	36 months experience with 1 session per week	Last session was 17 months prior
	DJ	Yes	Positive	1 month experience with 2 sessions per week	Last session was 1 month prior
	Dozer	No	N/A	N/A	N/A
1	Fury	No	N/A	N/A	N/A
	Gunner	No	N/A	N/A	N/A
	Osa	Yes	Negative	72 months experience with 1 session per week	Last session was 12 months prior
	Ross	No	N/A	N/A	N/A
	Sheridan	No	N/A	N/A	N/A
2	Ivey	Yes	Positive	12 months experience with 2 sessions per week	Last session was 6 months prior
	Toby	Yes	Positive	24 months experience with 2 sessions per week	Last session was within 1 week of the study starting

## Data Availability

The data presented in this study are openly available in OSF at DOI: 10.17605/OSF.IO/VK9YZ.

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
