# Peer review of "A Formalized Method to Acclimate Dogs to Voluntary Treadmill Locomotion at Various Speeds and Inclines"

_animals, 2022, doi:10.3390/ani12050567_

Round 1

Reviewer 1 Report

The review article about “ A Formalized Method to Acclimate Dogs to Voluntary Treadmill Locomotion at Various Speeds and Inclines” is of great interest for the amateur and professional sport training. Especially for trainers, owners and veterinary practitioners but also for the enthusiasts. The present paper is interesting. However, it needs several correction.

First of all the Authors should rewrite several paragraphs because sometime it is hard to follow. Also very often the jargon is used instead scientific language.

Introduction

It should be rewritten. In my opinion information about stress response after race and endurance training sessions in animals (ex. horses). It is important to show that exercise should be not stressful for dogs, as well as they should be acclimated with the treadmill. It is very important because welfare of the athletic animals is one of the most important issues in the sport industry.  

 In addition, the differences in movement biomechanics in treadmill and track should be emphasized. On the treadmill, the length of the gait is longer than for overground conditions. Moreover, different stiffness levels compared to a racetrack costs the animal different amounts of energy. Which was confirmed in race horses in thermographic evaluation.

I encourage to better understanding the more information about muscle stiffness examination should be added. In addition, the Latin names of examinated muscles should be incorporated.

I encourage not to use the names of dogs but numbers. It is more professional and in higher scientific soundness.

It will be very interesting if Authors will evaluate the hematological parameters which is commonly used in athletes such as RBC, Ht, Hb, WBC, LAC. Also heart rate should be evaluated. Measurements of cortisol as stress hormone will be perfect.

L364-366 – this was mentioned in previous parts. In horses there is no possibility to perform treadmill examination without acclimation.

The part Materials and Methods is very long. It should be more reader friendly.

Author Response

Thank you for reviewing the manuscript and for your kind and constructive comments. 

We have reviewed the manuscript and made every attempt to eliminate any jargon to ensure a balance between readability and technical precision. We recognize that our audience is broad and we do want this protocol to be widely adopted. Therefore, we have elected to keep some of this common language. Please see these lines changed: L90, L119, L141-144, L156, L400-404.

In reference to your comment about the stress response after training or racing, we agree exercise training for dogs should not be stressful. However, from our research we have seen some research studies do use forced exercise protocols. In addition, the purpose of this paper is to provide a detailed method to behaviorally acclimate dogs to the treadmill in the most low-stress method possible. This will allow a handler to read this paper and use it to acclimate their dog using low-stress methods. 

L36-37: Thank you for your comment on the biomechanical differences between running on the treadmill and running on flat ground. This a good point to remind our readers, and as such have included a statement to that effect on L36-37. The goal of the manuscript, however, is not to review the benefits or downsides of using a treadmill, rather if it is used, we want to make sure that the acclimation protocol is performed with minimal stress. 

L147-149: Thank you for your comment on the muscle soreness examination. The muscle soreness evaluation is a simple approach that does not require veterinary training, thus making it applicable to individuals in the field. Thank you for your comment on using the latin names of the muscles palpated. We have realized we did not palpate specific muscles (except the iliopsoas), but muscle groups. We have made changes to our terminology on L147-149 to better reflect what was palpated. 

We appreciate the reviewer’s recommendation on referring to the dogs by numbers rather than by their names, we believe it is easier for the reader to comprehend the specific details of each dog by referring to them by their name rather than number. We will defer to the editor for their recommendation. 

L513-514: We agree it would be interesting to replicate the study and measure various biomarkers. The use of biomarkers (e.g. salivary cortisol, heart rate) to measure the stress in the dogs was beyond the scope of the paper, but we have added a statement to the future directions on L513-514. 

L364: Yes, we are restating the gap that needs to be filled. We agree horses should be acclimated, but we are referring to research studies which sometimes use forced exercise protocols or have used short acclimation protocols for the purpose of a brief examination, not the long-term acclimation of the horse to treadmill training. The purpose of this acclimation protocol was to use low-stress methods to prevent a dog’s aversion to the treadmill so as to utilize the treadmill as a long-term training method. 

One purpose of this paper is to create a document that is relatively easy to read for the layperson (so they can use this for their dog). As a methods paper, we have made the methods section explicit and with sufficient detail for anyone to reproduce the methodology. We have provided a more succinct protocol document in the supplemental information, but felt that the details are necessary.

Reviewer 2 Report

Article “A Formalized Method to Acclimate Dogs to Voluntary Tread- 2 mill Locomotion at Various Speeds and Inclines” from Sigall, Ramos and Otto researchers:

Although performed over a small number of dogs the present work contributes to provide specific details on acclimation protocols, which is of the outmost importance for scientific community. Levels of difficulties, timeline, number of sessions, scheduling and type of reward are described and it also takes in consideration previous familiarization to treadmill and evaluation of previous orthopaedic conditions (in order to exclude the possibilities of re-injuries of exacerbation of sub-clinical conditions) as it also crosses the sessions results with gait analysis and evaluation of muscle soreness.

Note. On page 11 line 404 the sentence “Thirteen of the 70 acclimation 402 sessions ended without completing a new level”, must be contextualized or explained as being the total  number of sessions considering all the dogs. In fact,  it appears somewhat “loose” as it was stated before that he protocol was efficient with a mean of 9.4 sessions (+/- 1.9) required to fully acclimate a dog.

Limitations concerning this work are small number of dogs considering the great possibilities of morphological typologies of these species.  Altogether, the work is a nice contribution to define clear protocol parameters to acclimate dogs at various speeds and inclines treadmill exercises, allowing others to work over these issues with confidence.

Author Response

Thank you so much for reviewing the manuscript and providing your kind comments. We firmly believe this paper should add these details to make the study repeatable and easy for a layperson to understand and use for acclimating their personal dog. 

Formerly L402; Currently L400-404: We agree this sentence needs to be refined. We have contextualized the statement and provided further detail for the reader’s understanding of the data.

Round 2

Reviewer 1 Report

The Authors fullfilled my concerns. The article may be published in this shape.